# Postpartum haemorrhage occurring in UK midwifery units: A national population-based case-control study to investigate incidence, risk factors and outcomes

**Madeline Elkington**[1]*, **Jennifer J. Kurinczuk**[1], **Dharmintra Pasupathy**[2], **Rachel Plachcinski**[3], **Jane Rogers**[4], **Catherine Williams**[5], **Rachel Rowe**[1], on behalf of the UKMidSS Steering Group[¶]

1 NIHR Policy Research Unit in Maternal and Neonatal Health and Care, National Perinatal Epidemiology Unit, Nuffield Department of Population Health, University of Oxford, Oxford, United Kingdom,
2 Reproduction and Perinatal Centre, Faculty of Medicine and Health, University of Sydney, Sydney, Australia, 3 Independent Parent, Patient and Public Involvement Consultant, Dewsbury, United Kingdom,
4 Consultant Midwife, Formerly at University Hospitals Southampton, Southampton, United Kingdom,
5 Independent Parent, Patient and Public Involvement Consultant, Henley on Thames, United Kingdom

¶ Membership of the UKMidSS Steering Group is provided in the Acknowledgments
* madeline.elkington@dph.ox.ac.uk

**Data Availability Statement:** The UKMidSS PPH dataset used for this study cannot be made publicly available because it contains information which

## Abstract

### Objectives

To estimate the incidence of, and investigate risk factors for, postpartum haemorrhage (PPH) requiring transfer to obstetric care following birth in midwifery units (MU) in the UK; to describe outcomes for women who experience PPH requiring transfer to obstetric care.

### Methods

We conducted a national population-based case-control study in all MUs in the UK using the UK Midwifery Study System (UKMidSS). Between September 2019 and February 2020, 1501 women with PPH requiring transfer to obstetric care following birth in an MU, and 1475 control women were identified. We used multivariable logistic regression, generating adjusted odds ratios (aORs) and 95% confidence intervals (CIs) to investigate risk factors for PPH requiring transfer to obstetric care.

### Results

The incidence of PPH requiring transfer to obstetric care following birth in an MU was 3.7% (95% CI 3.6%-3.9%). Factors independently associated with PPH requiring transfer to obstetric care were smoking during pregnancy (aOR = 0.73; 95% CI 0.56–0.94), nulliparity (aOR = 1.96; 95% CI 1.66–2.30), previous PPH (aOR = 2.67; 95% CI 1.67–4.25), complications in a previous pregnancy other than PPH (aOR = 2.40; 95% CI 1.25–4.60), gestational age ≥41 weeks (aOR = 1.36; 95% CI 1.10–1.69), instrumental birth (aOR = 2.69; 95% CI 1.53–4.72), third stage of labour ≥60 minutes (aOR = 5.56; 95% CI 3.93–7.88), perineal

could identify participating centres, raising confidentiality issues. Requests for access to the dataset underlying our findings will be considered by the National Perinatal Epidemiology Unit Data Sharing Committee and should be addressed to ukmidss@npeu.ox.ac.uk in the first instance.

**Funding:** This paper presents independent research funded by the National Institute for Health Research (NIHR) (https://www.nihr.ac.uk/) under its Research for Patient Benefit (RfPB) Programme (PB-PG-0418-20005) (JJK, DP, RP, JR, CW, RR) and conducted through the Policy Research Unit in Maternal and Neonatal Health and Care, PR-PRU-1217-21202 (JJK, RP, RR). ME was funded by a University of Oxford Nuffield Department of Population Health DPhil Scholarship (https://www.ndph.ox.ac.uk/). The views expressed are those the author(s) and not necessarily those of the NIHR or the Department of Health and Social Care. The funders had no role in study design, data collection and analysis, decision to publish, or preparation of the manuscript.

**Competing interests:** The authors have declared that no competing interests exist.

trauma (aOR = 4.67; 95% CI 3.16–6.90), and birthweight 3500-3999g (aOR = 1.71; 95% CI 1.42–2.07) or $\geq$4000g (aOR = 2.31; 95% CI 1.78–3.00). One in ten (10.6%) cases received a blood transfusion and one in five (21.0%) were admitted to higher level care.

## Conclusions

The risk factors identified in this study align with those identified in previous research and with current guidelines for women planning birth in an MU in the UK. Maternal outcomes after PPH were broadly reassuring and indicative of appropriate management. NHS organisations should ensure that robust guidelines are in place to support management of PPH in MUs.

## Introduction

Childbirth in the UK is generally safe for women and their babies, and complications following birth are relatively rare, prticularly for women who are healthy with straightforward pregnancies. Most women who give birth in the UK do so in a consultant-led obstetric unit (OU), however, around 13% of births occur in midwifery units (MUs) [1]. Since the early 1990s, support for women's choices, specifically around place of birth, has been a central focus of UK maternity care policy [2]. Since 2014, national guidance has explicitly recommended that women at low risk of complications should be able to choose between birth at home, in an MU or in an OU [3]. In the UK, MUs may be either located on the same site as an OU, referred to as 'alongside' midwifery units (AMUs), or in a separate location from an OU, freestanding midwifery units (FMUs). Care in MUs is provided by midwives, and transfer to an OU is required for obstetric or medical care [3]. In an AMU, this may involve moving to another ward or floor in the same hospital, or, in some cases, the woman may remain in the AMU while receiving care from an obstetrician. Transfer from an FMU typically takes place by ambulance. The current national guideline suggests that planning birth in an MU is particularly suitable for women who are healthy with straightforward pregnancies because it is associated with a lower risk of intervention and no increased risk of adverse outcome for mothers or babies [3].

Many of the risk factors for postpartum haemorrhage (PPH), excessive bleeding after childbirth, including previous caesarean section [4], multiple gestation [5] and hypertension [6, 7] are less common in women who plan birth in an MU because women with these risk factors are generally advised to plan birth in an OU [3]. Admission criteria vary between MUs, however. A survey of admission criteria for MUs found that most guidelines (86%) listed at least one criterion that was considered 'more inclusive' than the national guidelines, that is, admitting women who have one or more risk factor identified in the NICE guidelines [8]. Other research using data from UK MUs confirms that women who might be considered to be at a higher risk of complications, including those with a pre-existing medical risk factor or pregnancy complication, are admitted to MUs, albeit in relatively small numbers [9, 10]. Women at low risk of complications who plan birth in MUs are less likely to develop complications during labour and birth, including PPH, compared with women at low risk of complications who plan birth in an OU [11]. However, women admitted to MUs may still experience a PPH; some after transfer to obstetric care and some following birth in an MU. In an MU setting, there may be a delay in access to medical treatments for PPH such as blood products or an operating theatre, particularly in FMUs, where a transfer for medical care necessarily involves travel to a different location. To ensure births in an MU are as safe as possible, recognition of

women who may be at increased risk of a PPH and the prompt diagnosis and management of a PPH is critical.

There is currently limited evidence about how often PPH occurs in MUs, the risk factors for PPH among women who give birth in MUs or outcomes for women who experience PPH in MUs. Such evidence would help inform decision-making for women considering or planning birth in an MU and the health professionals providing their care.

This study aimed to (a) estimate the incidence of PPH requiring transfer to obstetric care following birth in an MU in the UK, (b) investigate the risk factors for PPH requiring transfer to obstetric care among women who give birth in MUs in the UK, and (c) identify risk factors for admission to higher level care or blood transfusion, referred to below as 'enhanced treatment or care'.

## Methods

### Study design

We carried out a national, population-based, case-control study.

### Cases and controls

Cases were identified as all women who gave birth in an MU in the UK between 1 September 2019 and 29 February 2020, who experienced a PPH requiring transfer to obstetric care, where the primary or secondary reason for transfer was PPH. One control per case was identified as the woman who did not meet the case definition who gave birth in the same MU immediately before each case.

### Data collection

We collected anonymised information from MUs using the UK Midwifery Study System (UKMidSS), a national research infrastructure involving all 127 AMUs and 79 FMUs in the UK at the time of the study. The UKMidSS infrastructure is described in detail elsewhere [12]; set up in 2015 to cover all UK AMUs, it was extended in 2019 to also involve all UK FMUs. In brief, UKMidSS comprises a network of midwife 'reporters' who respond to monthly email requests for data about numbers of admissions, births and 'cases' for UKMidSS studies. Reporters entered anonymised data for cases and controls from medical records using a secure web-based system. This study was intended to run for 12 months, however, due to the COVID-19 pandemic, and following guidance from the funder, active data collection was terminated after six months.

### Data and definitions

In the UK, at the time of data collection, blood loss volume was typically estimated visually [13], which is known to be inconsistent and inaccurate [14, 15]. We therefore used the case definition, 'PPH requiring transfer to obstetric care', rather than a specified blood loss volume, as a pragmatic indicator of more severe PPH. For women giving birth in FMUs, and for most women giving birth in AMUs, obstetric care for PPH would be provided in an OU, following physical transfer of the woman. In some circumstances, for some women giving birth in AMUs and experiencing PPH, an obstetrician might come into the AMU.

Maternal socio-demographic and clinical characteristics, pregnancy-related factors, intrapartum- and birth-related factors were considered as putative risk factors.

Socioeconomic status was derived from the woman's occupation using the three-class version of the National Statistics Socio-economic Classification (NS-SEC), using the 'simplified

method' [16]. Additional categories were created for 'employment status unknown', 'employed, but occupation unrecorded/uncodable' and 'unemployed/student'. Where a woman was unemployed or her employment status was unknown, the partner's occupation was used to derive socioeconomic status, if applicable. 'Area-based deprivation quintile' was derived using the woman's postcode, which UKMidSS reporters entered into a bespoke website that returned a 'score' for the Children in Low-income Families Local Measure. This score represents the proportion of children in the area aged under 16 living in households in receipt of out of work benefits, or tax credits where their reported income is less than 60% of UK median income [17].

Women were classified as having a previous pregnancy complication if any of the following were reported: previous PPH requiring transfer or treatment, previous Caesarean section, retained placental requiring manual removal, uterine surgery other than Caesarean section. We collected information about the following pre-existing medical conditions: essential hypertension, confirmed cardiac disease, thromboembolic disorder, atypical antibodies, hyperthyroidism, diabetes, renal disease, epilepsy and 'other' medical conditions. Information was also collected about the following current pregnancy factors: BMI at booking $>35kg/m^2$, post-term pregnancy ($>42$ weeks), anaemia, Group B Streptococcus, antepartum haemorrhage, pre-eclampsia/pregnancy induced hypertension, gestational diabetes, malpresentation, and 'other'. Maternal and fetal complications that, according to the national guidelines [3] may indicate the need for transfer to obstetric care, were also collected.

Data about the PPH, including estimated blood loss volume and cause and management were collected, as were neonatal (Apgar score at 5 minutes, neonatal admission to higher level care and neonatal morbidity) and maternal outcomes (admission to higher level care, maternal morbidity and blood transfusion). Among women who had a PPH requiring transfer to obstetric care, those who received a blood transfusion or were admitted to higher-level care were classified as receiving 'enhanced treatment or care'.

## Analysis

The incidence of PPH requiring transfer to obstetric care was estimated with 95% CIs, overall and in AMUs and FMUs separately, using the number of confirmed cases as the numerator and the total number of births in MUs (and in each type of MU) during the study period as the denominator. A two-sample test of proportions was conducted to compare the incidence observed in the two types of unit.

Data entry completeness was calculated for each unit type using the number of confirmed cases as the numerator and total number of cases reported minus the number of ineligible cases as the denominator. The incidence of PPH in each unit type was calculated using the total number of cases (with complete and incomplete data) in each unit type as the numerator and the number of births in each unit type as the denominator.

The characteristics of cases and controls were described using frequencies and proportions. Univariable unconditional logistic regression was used to investigate associations between potential explanatory variables and PPH requiring transfer to obstetric care, estimating unadjusted odds ratios (ORs) with 95% CIs. Robust variance estimation was used to allow for the clustering of women within MUs.

We conducted multivariable regression analysis generating adjusted ORs (aORs) with 95% confidence intervals, using a stepwise forward regression approach. Variables with a P-value $<0.1$ in the univariable analysis, or where there was evidence of confounding, were considered for inclusion in the multivariable model. Variables were entered into the regression equation from distal to proximal, with sociodemographic and pre-existing risk factors being entered

first, followed by pregnancy-related, intrapartum and birth-related factors. The impact of each variable as they were added was examined and assessed using the Wald test; those variables for which p<0.05 were retained in the model. We excluded 'Administration of Syntocinon/ Syntometrine for 3rd stage management' from the multivariable analysis, despite it being significant at the 0.05 level, because it was not possible to determine if the drug was administered in response to increased blood loss during the third stage.

Blood loss volume and maternal and neonatal outcomes among cases and controls were described using frequencies and percentages. Among cases, the causes of PPH and any 'enhanced treatment or care', as defined above, were tabulated using frequencies and percentages. Risk factors for 'enhanced treatment or care' among women who had a PPH requiring transfer were investigated using univariable logistic regression, generating ORs with 95% CIs.

We conducted two post hoc analyses to explore factors that might explain the difference in incidence of PPH in FMUs and AMUs. First we used frequencies and proportions and the Chi-square test to compare cases and controls giving birth in AMUs and FMUs using the risk factors identified as associated with PPH requiring transfer to obstetric care in the multivariable analysis. We also compared blood loss volume in cases and controls in women giving birth in different types of unit, using the same approach. Statistical significance was set at p <0.05.

## Small numbers and missing data

For privacy reasons, table cells with numbers smaller than five have been suppressed. Data completeness was high for most variables. However, data were not assumed to be missing at random and therefore a 'Missing' or 'Not recorded' category was created for all variables with ≥1% missing data. For the multivariable analysis, a complete case analysis was conducted for variables with <1% of missing data, meaning records with missing data for these variables were excluded from the analysis. For variables with ≥1% missing data, the 'Missing' category was included in the analysis and therefore records with missing data for these variables were retained in the analysis.

All analyses were conducted using Stata V.15.

## Sample size and power

The study was planned for a 12-month period with an anticipated incidence of PPH requiring transfer to obstetric care of 1% [18]. Based on an anticipated 107,000 births in MUs over 12 months, we estimated that the study would have 80% power at the 5% level of significance to detect ORs of 1.4 or greater and 1.7 or greater, for putative risk factors with a prevalence of 15% (e.g. gestational age >40 weeks) and 4% (e.g. current pregnancy complication), respectively. The actual number of cases and controls identified during the curtailed study period gave an estimated power of 80% at the 5% level of significance to detect ORs of 1.3 or greater and 1.6 or greater, assuming the same putative risk factors.

## Ethics

UKMidSS received approval from the National Research Ethics Service (NRES) Committee South West–Frenchay (REC ref. 15/SW/0166) in May 2015, and this study was approved as a substantial amendment to that approval (SA03) in July 2019. Because this study used anonymised data collected directly from participating units, consent from participants was not required.

## Results

### Response and incidence

All 206 MUs in the UK were invited to participate in the study and 200 units submitted at least one monthly report between September 2019 and February 2020 (97% of all UK MUs). The response to monthly report requests was 95%.

During the 6-month study period, UKMidSS midwives in participating units reported a total of 39,953 women who gave birth in MUs in the UK. There were 1,673 cases of PPH requiring transfer reported, with 1,501 confirmed cases and 1,475 confirmed controls after exclusion of ineligible cases/controls and duplicates (Fig 1). Based on confirmed cases, the overall incidence of PPH requiring transfer to obstetric care was 3.7% (95% CI 3.6–3.9).

The incidence of PPH requiring transfer was significantly higher in AMUs (3.9%; 95% CI 3.6–4.1) compared with FMUs (2.6%; 95% CI 2.1–3.2) (p <0.001). Because data entry completeness was higher in AMUs (91%) compared with FMUs (80%) we also estimated incidence based on reported cases, rather than confirmed cases, but this did not materially change our results: AMU incidence 4.3%; 95% CI 4.0–4.5; FMU incidence 3.2%; 95% CI 2.7–3.8.

### Postpartum blood loss

Almost all cases (95.6%) had an estimated blood loss >500mL (range: 400-5500mL; median: 1050mL; IQR 700-1450mL) while most controls (93.4%) had an estimated blood loss ≤500mL (range: 20-1300mL; median: 300mL IQR: 200-380mL) (Table 1, Fig 2). There were 92 controls (6.2%) who had an estimated blood loss ≥500mL (Table 1); PPH for these control women was managed by midwives in the MU without transfer to obstetric care.

### Risk factors for PPH requiring transfer

In the univariable analysis, the sociodemographic, pre-existing clinical and pregnancy-related factors significantly associated with PPH requiring transfer were: not smoking, socioeconomic

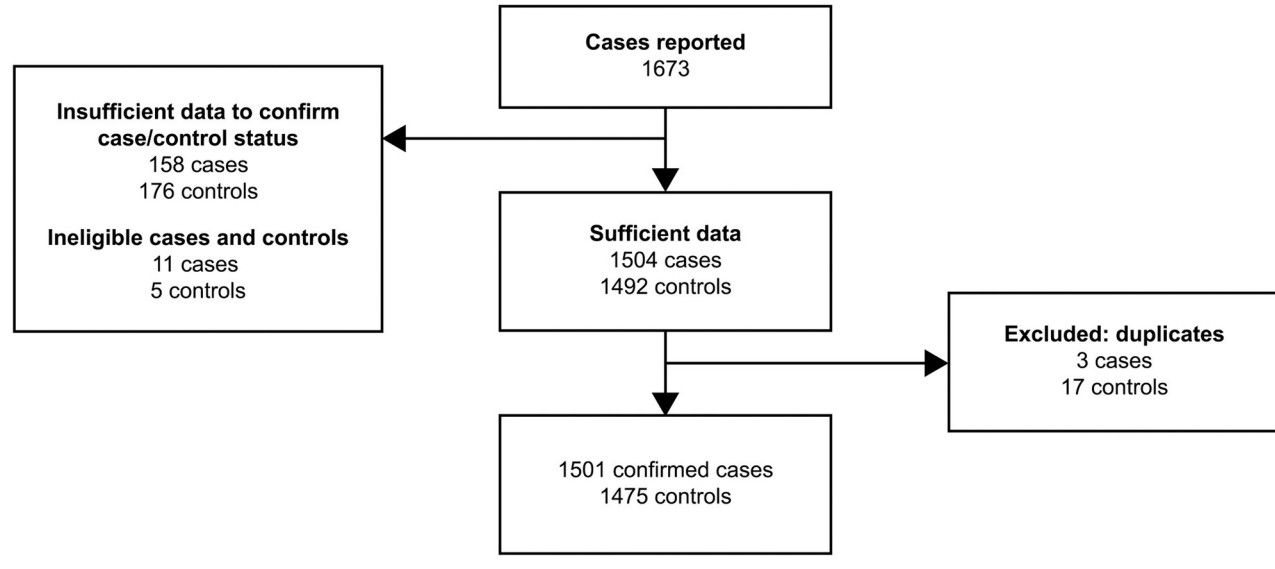

**Fig 1. Reported and confirmed cases.**

**Table 1. Postpartum blood loss among cases and controls.**

| | Controls n = 1475 | | Cases n = 1501 | |
|---|---|---|---|---|
| | n | % | n | % |
| **Blood loss (mL)** | | | | |
| <500 | 1377 | 93.4 | <5 | <0.3 |
| 500 | 46 | 3.1 | 61 | 4.1 |
| 501–999 | 43 | 2.9 | 554 | 36.9 |
| 1000–1499 | 3 | 0.2 | 517 | 34.4 |
| ≥1500 | 0 | 0.0 | 363 | 24.2 |
| Missing | 6 | 0.5 | <5 | <0.3 |

status, nulliparity, previous PPH, previous pregnancy complication other than PPH, and gestational age of 41 weeks' or more (Table 2).

The intrapartum and birth-related factors significantly associated with PPH requiring transfer in the univariable analysis were: induction of labour, immersion in water during labour, maternal complications identified at the start of labour care, fetal complications identified at the start of labour care, maternal complications identified during labour, instrumental vaginal delivery, a third stage of labour ≥60 minutes, perineal trauma, Syntocinon/ Syntometrine for 3rd stage management, and birthweight ≥3500g (Table 3).

Multivariable analysis indicated that, compared with controls, women who experienced a PPH requiring transfer to obstetric care were less likely to have smoked during pregnancy (aOR = 0.73; 95% CI 0.56–0.94) (Table 4). They were also more likely to be giving birth for the first time (aOR = 1.96; 95% CI 1.66–2.30); to have experienced a previous PPH (aOR = 2.67;

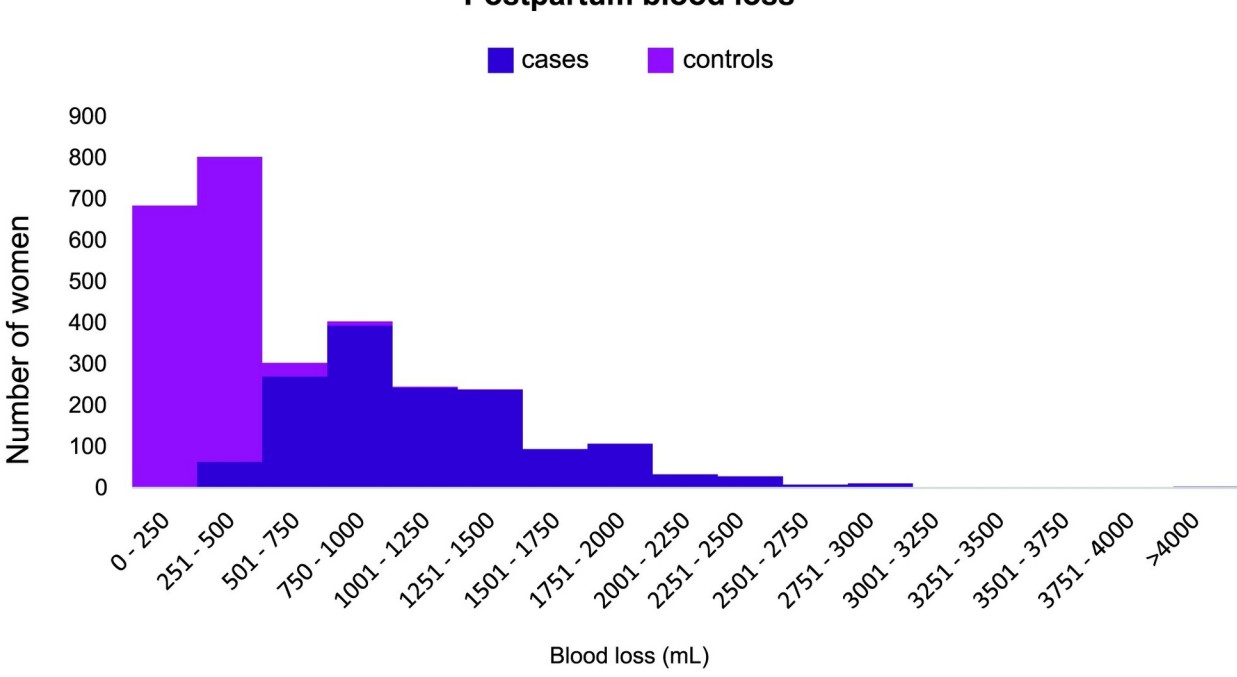

**Fig 2. Postpartum blood loss by case/control status.**

**Table 2. Sociodemographic, pre-existing and pregnancy-related characteristics of women.**

| | Controls (n = 1475) | | Cases (n = 1501) | | Unadjusted ORs | | p value |
|---|---|---|---|---|---|---|---|
| | n | % | n | % | OR | 95% CI | |
| **Maternal age** | | | | | | | 0.68 |
| <20 | 50 | 3.4 | 48 | 3.2 | 0.98 | (0.66–1.46) | |
| 20–24 | 216 | 14.6 | 205 | 13.7 | 0.97 | (0.75–1.25) | |
| 25–29 | 427 | 28.9 | 418 | 28.1 | 1 | . | |
| 30–34 | 498 | 33.8 | 552 | 36.7 | 1.13 | (0.93–1.38) | |
| 35–40 | 260 | 17.6 | 253 | 16.6 | 0.99 | (0.77–1.27) | |
| >40 | 24 | 1.6 | 25 | 1.6 | 1.06 | (0.61–1.85) | |
| Missing | 0 | . | 0 | . | . | . | |
| **Smoking status** | | | | | | | 0.005 |
| Did not smoke during pregnancy | 1274 | 8.4 | 1351 | 90.0 | 1 | . | |
| Smoked during pregnancy | 171 | 11.6 | 121 | 8.1 | 0.67 | (0.52–0.85) | |
| Missing | 30 | 2.0 | 29 | 1.9 | 0.91 | (0.62–1.34) | |
| **Area-based deprivation quintile** | | | | | | | 0.13 |
| 1st (least deprived) | 321 | 21.9 | 380 | 25.5 | 1 | . | |
| 2nd | 312 | 21.3 | 201 | 20.2 | 0.81 | (0.67–0.99) | |
| 3rd | 297 | 20.3 | 270 | 18.1 | 0.77 | (0.62–0.95) | |
| 4th | 276 | 18.8 | 274 | 18.4 | 0.84 | (0.68–1.03) | |
| 5th (most deprived) | 260 | 17.7 | 265 | 17.8 | 0.86 | (0.69–1.06) | |
| Missing | 10 | . | 11 | . | . | . | . |
| **Ethnicity** | | | | | | | 0.94 |
| White (UK and Ireland) | 986 | 66.9 | 997 | 66.4 | 1 | . | |
| White (other) | 182 | 12.3 | 189 | 12.6 | 1.03 | (0.82–1.29) | |
| Asian | 156 | 10.6 | 171 | 11.4 | 1.08 | (0.85–1.38) | |
| Black | 64 | 4.3 | 64 | 4.3 | 1.00 | (0.70–1.39) | |
| Other | 87 | 5.9 | 80 | 5.3 | 0.91 | (0.63–1.31) | |
| Missing | 0 | . | 0 | . | . | . | |
| **Socioeconomic status** | | | | | | | 0.010 |
| Higher managerial | 428 | 29.0 | 526 | 35.0 | 1 | . | |
| Intermediate | 289 | 19.6 | 264 | 17.6 | 0.75 | (0.61–0.91) | |
| Routine and manual | 358 | 24.3 | 335 | 22.3 | 0.76 | (0.64–0.91) | |
| Employed, occupation unknown | 104 | 7.1 | 103 | 6.9 | 0.81 | (0.60–1.08) | |
| Unemployed/ student | 101 | 6.9 | 80 | 5.3 | 0.65 | (0.47–0.90) | |
| Not recorded | 195 | 13.2 | 193 | 12.9 | 0.80 | (0.66–0.98) | |
| **BMI at booking** | | | | | | | 0.73 |
| < 18.5 | 48 | 3.3 | 42 | 2.8 | 0.88 | (0.57–1.37) | |
| 18.5–24.9 | 797 | 54.0 | 789 | 52.6 | 1 | . | |
| 25–29.9 | 406 | 27.5 | 432 | 28.8 | 1.07 | (0.91–1.27) | |
| 30–35 | 153 | 10.4 | 168 | 11.2 | 1.11 | (0.88–1.40) | |
| >35 | 28 | 1.9 | 30 | 2.0 | 1.08 | (0.62–1.88) | |
| BMI not recorded | 43 | 2.9 | 40 | 2.7 | 0.94 | (0.65–1.37) | |
| **Parityº** | | | | | | | < .001 |
| 0 | 513 | 34.8 | 750 | 50.0 | 1.74 | (1.51–2.02) | |
| 1 | 654 | 44.3 | 548 | 36.5 | 1 | . | |
| 2 or more | 308 | 20.9 | 203 | 13.5 | 0.79 | (0.64–0.97) | |
| Missing | 0 | . | 0 | . | . | . | |
| **Pre-existing medical risk factors\*** | | | | | | | 0.52 |

(*Continued*)

**Table 2.** (Continued)

| | Controls (n = 1475) | | Cases (n = 1501) | | Unadjusted ORs | | p value |
|---|---|---|---|---|---|---|---|
| | n | % | n | % | OR | 95% CI | |
| No pre-existing medical risk factors | 1453 | 98.5 | 1474 | 98.2 | 1 | . | |
| One or more | 22 | 1.5 | 27 | 1.8 | 1.21 | (0.67–2.17) | |
| Missing | 0 | . | 0 | . | . | . | |
| **Previous pregnancy complications** ** | | | | | | | <0.001 |
| No previous complication | 909 | 94.5 | 650 | 86.6 | 1 | . | |
| Previous PPH | 37 | 3.9 | 74 | 9.9 | 2.80 | (1.77–4.41) | |
| Previous complication other than PPH | 16 | 1.7 | 27 | 3.6 | 2.36 | (1.33–4.17) | |
| **Current pregnancy problems** º | | | | | | | 0.79 |
| None | 1382 | 93.7 | 1403 | 93.5 | 1 | . | |
| One or more | 93 | 6.3 | 98 | 6.5 | 1.04 | (0.77–1.39) | |
| Missing | 0 | . | 0 | . | . | . | |
| **Sex of baby** | | | | | | | 0.93 |
| Male | 710 | 48.3 | 717 | 47.9 | 1.01 | (0.88–1.18) | |
| Female | 761 | 51.7 | 781 | 52.1 | 1 | . | |
| Missing | 4 | . | 3 | . | . | . | |
| **Gestational age (weeks)** | | | | | | | <0.001 |
| 36–37 | 66 | 4.5 | 38 | 2.5 | 0.58 | (0.37–0.91) | |
| 38 | 167 | 11.3 | 143 | 9.4 | 0.86 | (0.67–1.09) | |
| 39 | 399 | 27.1 | 358 | 23.9 | 0.90 | (0.77–1.05) | |
| 40 | 591 | 40.1 | 589 | 39.2 | 1 | . | |
| 41–43 | 251 | 17.0 | 373 | 24.9 | 1.49 | (1.21–1.83) | |
| Missing | 1 | . | 0 | . | . | . | |

º Number of previous pregnancies carried to at least 24 completed weeks' gestation

* Hypertension, confirmed cardiac disease, thromboembolic disorder, atypical antibodies, hyperthyroidism, diabetes, renal disease, epilepsy.

**Includes all women with a PPH including those who also had another previous pregnancy problem. Previous complications were: Retained placenta requiring manual removal and Caesarean section. Excludes primiparous women.

† Retained placenta requiring manual removal, Caesarean section, Uterine surgery excluding Caesarean section and shoulder dystocia. Excludes primiparous women.

º BMI at booking >35, Post-term (>42 weeks) Anaemia, Group B Streptococcus, Antepartum haemorrhage, Pre-eclampsia/pregnancy-induced hypertension, Gestational diabetes, Malpresentation (breech or transverse lie), multiple pregnancy

95% CI 1.67–4.25) or another complication in a previous pregnancy if they had given birth before (aOR = 2.40; 95% CI 1.25–4.60); to be giving birth at or after 41 weeks' gestation (aOR = 1.36; 95% CI 1.10–1.69). In term of risk factors related to the birth, women who had a PPH requiring transfer to obstetric care were more likely to have an instrumental birth (aOR = 2.69 95% CI 1.53–4.72); to have a third stage of labour lasting 60 minutes or longer (aOR = 5.56 95% CI 3.93–7.88); to experience a third or fourth degree perineal tear (aOR = 4.67 95% CI 3.16–6.90); or to give birth to a baby weighing 3500-3999g (aOR = 1.71 95% CI 1.42–2.07), or >4000g (aOR = 2.31 95% CI 1.78–3.00).

## Maternal and neonatal outcomes following a PPH

Overall, 158 cases (11%) received a transfusion of blood or blood products following their PPH (Table 5). One in five cases (21%) was admitted to higher level care, compared with <0.3% of controls. Five cases and no controls were admitted to intensive care. Of those admitted to higher level care, less than 2% of cases and no controls stayed longer than two days in

**Table 3. Intrapartum and birth-related factors.**

| | Controls (n = 1475) | | Cases (n = 1501) | | Unadjusted ORs | | p value |
|---|---|---|---|---|---|---|---|
| | n | % | n | % | OR | 95% CI | |
| **Stage of labour at start of care** | | | | | | | 0.63 |
| Latent stage | 276 | 18.7 | 270 | 18.1 | 0.94 | (0.77–1.15) | |
| Active 1st stage | 1030 | 70.0 | 1072 | 71.7 | 1 | . | |
| Passive 2nd stage | 53 | 3.6 | 46 | 3.1 | 0.83 | (0.58–1.20) | |
| Active 2nd stage | 113 | 7.7 | 108 | 7.2 | 0.92 | (0.69–1.22) | |
| Missing | 3 | . | 5 | . | . | . | |
| **Induction of labour** | | | | | | | 0.022 |
| No | 1412 | 96.2 | 1410 | 94.4 | 1 | . | |
| Yes | 56 | 3.8 | 84 | 5.6 | 1.50 | (1.06–2.13) | |
| Missing | 7 | . | 7 | . | . | . | |
| **Maternal complications identified at the start of labour care*** | | | | | | | 0.003 |
| None | 1455 | 98.9 | 1457 | 97.5 | 1 | . | |
| One or more | 16 | 1.1 | 38 | 2.5 | 2.37 | (1.33–4.23) | |
| Missing | 4 | . | 6 | . | | | |
| **Fetal complications identified at the start of labour care†** | | | | | | | 0.008 |
| None | 1443 | 98.1 | 1446 | 96.7 | 1 | . | |
| One or more | 28 | 1.9 | 49 | 3.3 | 1.75 | (1.15–2.64) | |
| Missing | 4 | . | 6 | . | . | . | |
| **Maternal complications identified during labour (before birth)°** | | | | | | | 0.007 |
| None | 1451 | 98.4 | 1458 | 97.2 | 1 | . | |
| One or more | 24 | 1.6 | 43 | 2.8 | 1.78 | (1.17–2.71) | |
| Missing | 0 | | 0 | | | | |
| **Fetal complications identified during labour (before birth)¶** | | | | | | | 0.005 |
| None | 1376 | 93.2 | 1360 | 90.6 | 1 | | |
| One or more | 99 | 6.8 | 141 | 9.4 | 1.44 | (1.12–1.86) | |
| Missing | 0 | | 0 | | | | |
| **Labour/birth in water** | | | | | | | 0.004 |
| No immersion in water | 820 | 55.7 | 808 | 54.0 | 1 | . | |
| Immersion in water for labour, land birth | 217 | 14.7 | 263 | 17.6 | 1.23 | (1.04–1.45) | |
| Birth in water | 435 | 29.6 | 425 | 28.4 | 0.99 | (0.86–1.14) | |
| Missing | 3 | . | 5 | . | . | . | |
| **Birth mode** | | | | | | | <0.001 |
| Spontaneous vertex birth | 1459 | 99.2 | 1456 | 97.7 | 1 | . | |
| Vaginal breech | <5 | <0.3 | 0 | 0 | . | . | |
| Instrumental | 10 | 0.7 | 35 | 2.4 | 3.51 | (2.25–5.43) | |
| Missing | 4 | . | 10 | . | . | . | |
| **Duration of third stage of labour** | | | | | | | <0.001 |
| < 60 minutes | 1399 | 94.9 | 1240 | 82.6 | 1 | . | |
| ≥ 60 minutes | 53 | 3.6 | 227 | 15.1 | 4.83 | (3.43–6.79) | |
| Missing | 23 | 1.6 | 34 | 2.3 | . | . | |
| **Perineal trauma** | | | | | | | <0.001 |
| <3rd degree tear or no tear | 1427 | 96.9 | 1290 | 86.2 | 1 | . | |
| 3rd or 4th degree tear | 45 | 3.1 | 207 | 13.8 | 5.09 | (3.49–7.42) | |
| Missing | 3 | . | 4 | . | . | . | |
| **Syntocinon/ Syntometrine for 3rd stage management** | | | | | | | 0.001 |
| No | 278 | 18.9 | 205 | 13.7 | 1 | | |

*(Continued)*

**Table 3.** (Continued)

| | Controls (n = 1475) | | Cases (n = 1501) | | Unadjusted ORs | | p value |
|---|---|---|---|---|---|---|---|
| | n | % | n | % | OR | 95% CI | |
| Yes | 1194 | 81.1 | 1291 | 86.3 | 1.47 | (1.17–1.83) | |
| Missing | 4 | . | 5 | . | . | . | |
| **Birthweight (g)** | | | | | | | <0.001 |
| <3000 | 192 | 13.1 | 112 | 7.5 | 0.74 | (0.57–0.96) | |
| 3000–3499 | 644 | 43.8 | 508 | 33.9 | 1 | . | |
| 3500–3999 | 485 | 33.0 | 628 | 41.9 | 1.64 | (1.37–1.96) | |
| ≥4000 | 150 | 10.2 | 250 | 16.7 | 2.11 | (1.66–2.69) | |
| Missing | 4 | . | 3 | . | . | . | |

\* Maternal tachycardia, hypertension, proteinuria, maternal pyrexia, vaginal blood loss, prolonged rupture of membranes, and reported pain differing from pain normally associated with contractions.

† Significant meconium, abnormal presentation, high or free-floating head, suspected anhydramnios or polyhydramnios, fetal heart rate abnormality, deceleration in fetal heart rate, and reduced fetal movements in the last 24 hours.

º Maternal tachycardia, hypertension, maternal prexia, vaginal blood loss, prolonged rupture of membranes.

⁵ Significant meconium, confirmed/suspected delay in first stage of labour, confirmed\suspected delay in second stage of labour, obstetric emergency, abnormal presentation, fetal heart rate abnormality, and deceleration in fetal heart rate.

higher level care. Initiation of breastfeeding before discharge was similar among cases (82%) and controls (80%).

One in four cases (25%) received 'enhanced treatment or care', compared with 0.6% of controls. Most women who received 'enhanced treatment or care' (79%) were admitted for higher level care, primarily for observation following PPH. There were no maternal deaths among cases or controls in this study.

Among women who had a PPH requiring transfer, those who received 'enhanced treatment or care' were not significantly different from those who did not in terms of maternal sociodemographic characteristics, pre-existing clinical risk factors or pregnancy-related factors (S1–S5 Tables).

The only labour- and birth-related factor associated with an increased risk of 'enhanced treatment or care' was duration of third stage of labour lasting 60 minutes or longer (OR 1.14 95% CI 1.34–2.41) (Table 6). Postpartum blood loss of 1000-1499mL was associated with higher odds of receiving 'enhanced treatment or care' (OR 5.53; 95% CI 1.70–17.79), as was blood loss of 1500mL or more (OR 33.0 95% CI 10.15–107.24), compared with blood loss of 500mL or less. Genital tract trauma as the primary cause of the PPH was associated with lower odds of 'enhanced treatment or care' (OR 0.58 95% CI 0.43–0.79).

Neonatal outcomes were generally good, with small numbers of babies reported as having a neonatal morbidity, Apgar score <7 at 5 minutes or admission to higher level care. These neonatal outcomes were slightly more common among babies born to women who had a PPH requiring transfer to obstetric care (S6 Table). Two babies died in the neonatal/post-neonatal period. There were no stillbirths in this study.

## Prevalence of risk factors among women giving birth in AMUs and FMUs

The prevalence of risk factors for PPH requiring transfer among cases and controls was similar in AMUs and FMUs (S7 and S8 Tables). There was no statistically significant difference between FMUs and AMUs in terms of blood loss volume among cases or controls (S9 Table).

## Discussion

This study provides valuable information about the incidence of and risk factors for PPH occurring in MUs in the UK, to support midwifery practice and women's decision-making. Among women giving birth in UK MUs, the estimated incidence of PPH requiring transfer to obstetric care was 3.7%. Several independent risk factors for PPH requiring transfer were identified through multivariable analysis. Of the risk factors that are known prior to admission for the birth, primiparity, not smoking in pregnancy, previous PPH, problems in a previous pregnancy other than PPH, and gestational age ≥41 weeks were associated with higher odds of having a PPH requiring transfer to obstetric care. The factors occurring during labour and birth associated with higher odds of having a PPH requiring transfer were instrumental birth, a third stage of labour lasting ≥60 minutes, 3rd or 4th perineal tear, and birthweight ≥3500g.

This study also provides information about outcomes following a PPH in an MU in the UK. Among women who had a PPH requiring transfer to obstetric care, a third stage of labour lasting 60 minutes or longer was associated with higher odds of requiring 'enhanced treatment or care', while genital tract trauma was associated with lower odds of requiring 'enhanced treatment or care'.

The incidence of PPH requiring transfer to obstetric care, following birth in a midwifery unit, was higher than we anticipated. There are, however, few data against which to compare our estimates. In the Birthplace national prospective cohort study, which investigated outcomes by planned place of birth in England in 2008–10, the proportion of women who were transferred from an MU to obstetric care where the primary reason for transfer was PPH was 1.0% in FMUs and 1.0% in AMUs [11]. However, these results may not be directly comparable as they only included women whose **primary** reason for transfer was PPH, whereas our study also included women for whom PPH was a secondary reason for transfer. The Birthplace study was carried out in 2008–10. Since then, the overall proportion of women experiencing a PPH in the UK has also increased, as it has in other high-income countries [6, 19, 20]. Between 2010 and 2021, the rate of PPH among spontaneous vaginal births in England more than doubled, from 7.2% to 16.0% [21, 22]. While these data relate to births in all NHS hospitals and thus are not directly comparable to births solely in MUs, it is possible that this general upward trend may also be seen in women giving birth in an MU. Research into the incidence of PPH in other high-income countries suggests that the recent increased incidence does not appear to be entirely explained by increasing prevalence of risk factors in the population [6, 19, 20, 23], which suggests that the incidence of PPH may have also increased among 'lower risk' women who give birth in MUs.

The incidence of PPH requiring transfer was higher in AMUs (3.9%) compared with FMUs (2.6%). Typically, women planning birth in FMUs have fewer pre-existing and current pregnancy complications compared with women who plan birth in an AMU [11], but our further analysis comparing the characteristics of women giving birth in the two types of unit indicated that this is unlikely to explain the difference in incidence we found. Our case definition for this study was 'PPH requiring transfer', rather than PPH, so this is likely to have been influenced by midwives' decision-making or 'threshold' for transfer. It is also possible therefore that the higher incidence of PPH requiring transfer in AMUs, compared with FMUs, may be explained by more readily-available access to obstetric care in AMUs. Midwives in AMUs might, as a consequence, have a lower 'threshold' for transfer for PPH, particularly for women whose blood loss might be managed under midwifery care, i.e. those with smaller amounts of blood loss. However, our post hoc analysis revealed that postpartum blood loss volume was similar in FMUs and AMUs among both cases and controls, suggesting that there were not significant numbers of women whose PPH was managed without transfer in FMUs.

**Table 4. Independent factors associated with PPH in midwifery units requiring transfer to obstetric care.**

| | Controls | | Cases | | Unadjusted analysis | | Adjusted analysis n = 2940 | | |
|---|---|---|---|---|---|---|---|---|---|
| | n | % | n | % | OR | (95% CI) | aOR | (95% CI) | p value |
| **Smoking status** | | | | | | | | | 0.031 |
| Did not smoke during pregnancy | 1274 | 86.4 | 1351 | 90.0 | 1 | . | 1 | | |
| Smoked during pregnancy | 171 | 11.6 | 121 | 8.1 | 0.66 | (0.52–0.85) | 0.73 | (0.56–0.94) | |
| Missing | 30 | 2.0 | 29 | 1.9 | 0.91 | (0.62–1.34) | 0.80 | (0.48–1.31) | |
| **Area-based deprivation quintile** | | | | | | | | | 0.019 |
| 1st (least deprived) | 321 | 21.9 | 380 | 25.5 | 1 | | 1 | | |
| 2nd | 312 | 21.3 | 201 | 20.2 | 0.81 | (0.67–0.88) | 0.87 | (0.71–1.07) | |
| 3rd | 297 | 20.3 | 270 | 18.1 | 0.77 | (0.62–0.95) | 0.88 | (0.71–1.09) | |
| 4th | 276 | 18.8 | 274 | 18.4 | 0.84 | (0.68–1.03) | 1.04 | (0.83–1.31) | |
| 5th (most deprived) | 260 | 17.7 | 265 | 17.8 | 0.86 | (0.69–1.07) | 1.27 | (1.00–1.61) | |
| **Parity** | | | | | | | | | <0.001 |
| 0 | 513 | 34.8 | 750 | 50.0 | 1.87 | (1.62–2.15) | 1.96 | (1.66–2.30) | |
| 1 | 654 | 44.3 | 548 | 36.5 | 1 | . | 1 | | |
| 2+ | 308 | 20.8 | 203 | 13.5 | 0.78 | (0.64–0.97) | 0.81 | (0.65–1.01) | |
| **Previous pregnancy complication** * | | | | | | | | | |
| No previous complication | 909 | 94.5 | 650 | 86.6 | 1 | . | 1 | . | |
| Previous PPH | 37 | 3.9 | 74 | 9.9 | 2.80 | (1.77–4.41) | 2.67 | (1.67–4.25) | <0.001 |
| Previous complication other than PPH | 16 | 1.7 | 27 | 3.6 | 2.36 | (1.33–4.17) | 2.40 | (1.25–4.60) | 0.009 |
| **Gestational age** | | | | | | | | | 0.007 |
| 36–37 | 66 | 4.5 | 38 | 2.5 | 0.58 | (0.37–0.91) | 0.69 | (0.42–1.14) | |
| 38 | 167 | 11.3 | 143 | 9.4 | 0.86 | (0.68–1.10) | 1.04 | (0.80–1.36) | |
| 39 | 399 | 27.1 | 358 | 23.9 | 0.90 | (0.77–1.05) | 0.98 | (0.82–1.17) | |
| 40 | 591 | 40.1 | 589 | 39.2 | 1 | . | 1 | . | |
| 41–43 | 251 | 17.0 | 373 | 24.9 | 1.49 | (1.21–1.84) | 1.36 | (1.10–1.69) | |
| **Birth mode** | | | | | | | | | <0.001 |
| Spontaneous vertex or vaginal breech birth | 1461 | 99.2 | 1456 | 97.7 | 1 | . | 1 | . | |
| Instrumental birth | 10 | 0.82 | 35 | 2.35 | 2.92 | (1.83–4.64) | 2.69 | (1.53–4.72) | |
| **Duration of third stage of labour** | | | | | | | | | <0.001 |
| < 60 minutes | 1399 | 94.9 | 1240 | 82.6 | 1 | . | 1 | . | |
| ≥ 60 minutes | 53 | 3.6 | 227 | 15.1 | 4.8 | (3.42–6.81) | 5.56 | (3.93–7.88) | |
| Missing | 23 | 1.6 | 34 | 2.3 | . | . | . | . | |
| **Perineal tear** | | | | | | | | | <0.001 |
| <3rd degree tear or no tear | 1427 | 96.9 | 1290 | 86.2 | 1 | . | 1 | . | |
| 3rd or 4th degree tear | 45 | 3.1 | 207 | 13.8 | 5.09 | (3.49–7.42) | 4.67 | (3.16–6.90) | |
| **Birthweight (gm)** | | | | | | | | | <0.001 |
| <3000 | 192 | 13.1 | 112 | 7.5 | 0.47 | (0.57–0.96) | 0.78 | (0.57–1.06) | |
| 3000–3499 | 644 | 43.8 | 508 | 33.9 | 1 | . | 1 | . | |
| 3500–3999 | 485 | 33.0 | 628 | 41.9 | 1.64 | (1.37–1.96) | 1.71 | (1.42–2.07) | |
| ≥4000 | 150 | 10.2 | 250 | 16.7 | 2.11 | (1.66–2.69) | 2.31 | (1.78–3.00) | |

º Each variable in the model is adjusted for all other variables in the model. Includes 2,940 observations

* Excludes primiparous women

**Table 5. Maternal outcomes among cases and controls.**

| | Controls n = 1475 | | Cases n = 1501 | |
|---|---|---|---|---|
| | **n** | **%** | **n** | **%** |
| **Blood transfusion** | na | na | 158 | 10.6 |
| **Maternal morbidity reported*** | <5 | <0.3 | 30 | 2.0 |
| **Admission to higher level of care** | <5 | <0.3 | 308 | 21.0 |
| **Type of higher level care†** | | | | |
| Enhanced maternity care | <5 | 100.0 | 303 | 98.3 |
| Intensive care | 0 | 0 | 5 | 1.6 |
| **Duration of stay in higher level care†** | | | | |
| < 1 day | 0 | 0 | <5 | <1.6 |
| 1 day | <5 | na | 269 | 88.8 |
| 2 days | <5 | na | 26 | 8.6 |
| 3 or more days | 0 | 0 | <5 | <1.6 |
| Missing | 0 | 0 | 5 | . |
| **Primary reason for higher level care†** | | | | |
| PPH and observation following PPH | 0 | 0.0 | 232 | 74.3 |
| Observation (without PPH specified) | 0 | 0 | 42 | 14.0 |
| Recovery from theatre procedure | <5 | <0.3 | 12 | 3.6 |
| Sepsis/pyrexia & Other/not clear | <5 | <0.3 | 22 | 7.1 |
| **Any 'enhanced treatment or care' ⁋** | 7 | 0.6 | 370 | 24.7 |
| **Breastfeeding initiated before discharge** | | | | |
| Yes | 1173 | 79.7 | 1221 | 81.5 |
| No | 298 | 20.3 | 276 | 18.4 |
| Missing | 4 | . | 4 | . |

* Maternal morbidity includes: readmission for secondary PPH/retained placenta, pyrexia/sepsis, anaemia and 'other' morbidities

† Proportions are among women admitted to higher level care

⁋ Presence of either of the following: blood transfusion, admission to higher level-care

The risk factors for PPH identified in this study broadly align with the few previous studies of PPH in populations of women considered at low risk of complications [24–26]. Smoking status was significantly but inversely associated with PPH requiring transfer to obstetric care, with women who smoked during pregnancy being almost 30% less likely to experience a PPH requiring transfer than those who did not smoke. The hypercoagulation effects of smoking may contribute to this association [27], but it is also possible that the observed association between not smoking and PPH requiring transfer is a result of residual confounding in our study. The literature about this association is inconclusive, with some studies reporting that smoking may be a protective factor for PPH [28–30] and others indicating it might be a risk factor [31, 32] suggesting that further research is required to investigate this association.

In contrast to some previous research [28, 33, 34], we did not find ethnicity to be significantly associated with PPH. A study using routine clinical data about more than 900,000 women giving birth in maternity units in England from 2015–17 found that women from ethnic minority backgrounds had an increased risk of severe PPH, after adjusting for some maternal, fetal and birth factors [34]. In our study, while the distribution of ethnicity overall was similar, the lack of association between ethnicity and PPH may be explained by the selected 'low risk' population giving birth in MUs.

**Table 6. Univariable analysis of risk factors for 'enhanced treatment or care' following PPH requiring transfer.**

| | No 'enhanced treatment or care' (n = 1,131) | | 'Enhanced treatment or care' (n = 370) | | Unadjusted ORs | | p value |
|---|---|---|---|---|---|---|---|
| | **n** | **%** | **n** | **%** | **OR** | **95% CI** | |
| **Duration of third stage of labour** | | | | | | | <0.001 |
| < 60 minutes | 956 | 84.5 | 284 | 76.8 | 1 | . | |
| ≥ 60 minutes | 148 | 13.1 | 79 | 21.4 | 1.79 | (1.34–2.41) | |
| Missing | 27 | 2.4 | 7 | 1.9 | 0.87 | (0.40–1.92) | |
| **Cause of PPH** | | | | | | | <0.001 |
| Uterine atony | 406 | 35,9 | 152 | 41.1 | 1 | . | |
| Genital tract trauma | 355 | 31.4 | 77 | 20.8 | 0.58 | (0.42–0.80) | |
| Retained products / morbidly adherent placenta | 221 | 19.5 | 106 | 28.7 | 1.28 | (0.95–1.73) | |
| Other | 15 | 1.3 | 6 | 1.6 | 1.07 | (0.45–2.52) | |
| Not recorded | 134 | 11.9 | 29 | 7.8 | 0.59 | (0.36–0.94) | |
| **Blood loss (mL)** | | | | | | | <0.001 |
| ≤500 | 60 | 5.3 | <5 | <1.2 | 0.91 | (0.24–3.42) | |
| 501–999 | 525 | 46.4 | 29 | 7.8 | 1 | . | |
| 1000–1499 | 405 | 35.8 | 112 | 30.3 | 5.01 | (3.06–8.18) | |
| 1500 | 137 | 12.1 | 226 | 61.1 | 33.0 | (17.14–52.02) | |
| Missing | 4 | . | <5 | <0.3 | . | . | |

Consistent with previous studies [33, 35, 36], we found strong evidence for nulliparity as a risk factor for PPH. Women who had not previously given birth were almost twice as likely to have a PPH requiring transfer, compared with women who had given birth once before. Women who give birth in MUs in the UK are more likely to have given birth before than be giving birth for the first time [11], and this was reflected in our study, with 65% of controls and 50% of cases being multiparous. While the risk of PPH among primiparous women in our study was higher than in women who had given birth before, the absolute risk of PPH following birth in a MU remains low for primiparous women giving birth in MU. Among all women giving birth in England, primiparous women have higher rates of PPH, compared with women who have given birth previously [22]. This, combined with the observation that primiparous women who had a PPH requiring transfer were not at higher risk of requiring 'enhanced treatment or care' compared with multiparous women, suggests that overall nulliparous women who are eligible to give birth in a MU should consider this option.

National guidance recommends that women who have had previous pregnancy complications, including a previous PPH requiring treatment or transfusion, should be advised to plan birth in obstetric units in the UK [3]. However, local NHS guidance about admission criteria for midwifery-led care varies widely from national guidance [8]. A national survey, carried out in 2018–19, found that in most MUs whose admission criteria explicitly mentioned PPH these criteria were not in alignment with current guidelines, with 1 in 4 (27%) admission guidelines that mentioned PPH as an admission criterion using previous blood loss <1L as the stated inclusion criteria, and 5% admitting women with a previous PPH <1.5L or <2L [8]. In our study, a small proportion of women had a previous PPH requiring treatment or transfusion (10% of cases and 4% of controls) and a smaller proportion had previous complications other than PPH (4% of cases and 2% of controls), including for example previous retained placenta requiring manual removal or previous caesarean section. Among women who had given birth

at least once before, previous PPH requiring treatment or blood transfusion increased a woman's odds of PPH requiring transfer by 2.7 times. Women who had a previous pregnancy complication other than PPH were 2.4 times more likely to have a PPH requiring transfer. Previous retained placenta requiring manual removal was the most common previous complication other than PPH, accounting for 70% of the cases with previous complications.

Several other intrapartum factors, including induction, immersion in water during labour, and complications identified during labour, were associated with increased odds of PPH requiring transfer at the univariable level, but not after adjustment for other factors. With the exception of immersion in water during labour, the proportion of women with these risk factors was relatively low, with very small numbers of women affected by each individual complication. Our study confirms advanced gestational age as an independent risk factor for PPH [31, 37–39]; women who gave birth at 41–42 weeks' were over 30% more likely to have a PPH requiring transfer, compared with women giving birth at 40 weeks' gestation.

In line with previous research, instrumental birth was associated with a 2.7 fold increase in the odds of PPH requiring transfer to obstetric care. Instrumental birth occurs only infrequently in MUs; in our study, 2.4% of cases and 0.7% of controls had an instrumental birth. It should be noted that this does not reflect the proportion of women who plan birth in a MU and have an instrumental birth, as women who were transferred from an MU to OU before the birth were not included in our study. Instrumental births in MUs are only performed in AMUs and almost always in circumstances in which expediting birth is a priority over physically transferring the woman to the obstetric unit. For women who do have an instrumental birth in a MU, our study suggests that vigilance by midwives following an instrumental birth is important.

The most significant independent risk factor for PPH requiring transfer was a third stage of labour lasting ≥60 minutes, which increased a woman's risk of PPH more than fivefold. This is consistent with previous research indicating that a longer third stage of labour is associated with an increased risk of PPH [40–42]. A prolonged third stage of labour can contribute to higher amounts of blood loss due to prolonged bleeding from the placental site and from unrepaired perineal trauma [43]. UK national guidance recommends that a third stage of labour should be diagnosed as prolonged if it lasts longer than 30 minutes with active management, or longer than 60 minutes with physiological management. Data were collected about the administration of syntocinon or syntometrine for active management of the third stage of labour, but with the available data it was not possible to determine whether syntocinon or syntometrine were administered as part of **planned** active management of labour or whether administration was indicated because of increased blood loss. Such information may have strengthened our analysis of the risk associated with duration of the third stage of labour.

Women who had a PPH requiring transfer were more likely to require 'enhanced treatment or care' (comprising admission to higher level care or blood transfusion) after birth, compared with controls (24.7% vs 0.6%). Among cases, women who required 'enhanced treatment or care' were not significantly different from women who did not, in terms of sociodemographic, clinical or pregnancy-related factors. This suggests it would not be feasible for midwives to identify women likely to be in need of enhanced care following PPH based on pre-identified characteristics. However, it is possible that the study was underpowered to detect differences in these potential risk factors because there were only small numbers of women in the 'enhanced treatment or care' group. The risk factors for enhanced treatment were more proximal labour- and PPH-related factors, including duration of the third stage of labour, cause of PPH and blood loss volume.

In our study, in a generally low risk population, 3.7% of births in midwifery units were affected by a PPH requiring transfer to obstetric care, with 1 in 4 cases having a reported blood

loss of 1500mL or greater. This represents significant potential for maternal morbidity. However, the broadly positive outcomes for women following PPH in an MU, evidenced in this study, appear indicative of appropriate management. The most common reason for admission to higher-level care was for observation following the PPH, which was indicated in some free text comments to be standard practice. Most women (90%) who were admitted to higher-level care stayed for less than two days, and only five (1.6%) were admitted to the ICU. Some pre-existing risk factors, including previous PPH and other previous pregnancy complications, were significantly associated with an increased risk of PPH requiring transfer, and there were indications at the univariable level that some complications arising during labour were also associated with an increased risk. There is no evidence however that planned birth in an OU or transfer to an OU prior to birth, for example for women with identified or emerging risk factors, would have either reduced the risk of having a PPH or improved outcomes following a PPH. There is also strong evidence that for women a positive birth experience includes clinical safety and psychosocial wellbeing, including involvement in decision-making, and care that is in line with their values and preferences[44], so women's choice, supported by appropriate evidence, is also important. In this context, to maintain safe care for women planning birth in MUs it is imperative that NHS organisations have robust guidelines about the management of PPH in MUs, appropriate equipment and training[45] and ready access to transfer when required.

## Strengths and limitations

A major strength of this study was its robust design; a national population-based case-control study which included all reported cases of PPH requiring transfer to obstetric care following birth in an MU. The inclusion of all MUs in the UK in the study and 98% participation rate minimises the risk of bias related to regional differences between MUs across the UK. Additionally, the high monthly reporting rate, with 95% response rate to monthly requests for data, reduced the likelihood of selection bias.

The case definition used in this study was based on the decision to transfer a woman to obstetric care, rather than on a specified volume of postpartum blood loss. This definition was chosen to capture women whose condition was considered severe enough to warrant transfer to obstetric care, rather than using estimated blood loss. At the time of data collection, visual estimation of blood loss was typically used in MUs, which previous studies have shown to be unreliable [14, 15], although quantitative measurement of blood loss is becoming more common [13]. Almost all cases had an estimated blood loss >500mL, the definition of PPH used in the UK [46], and almost all controls had an estimated blood loss ≤500mL. This suggests that the case definition employed here is comparable to other studies that use estimated blood loss of >500mL to define cases of PPH. The decision to transfer a woman for PPH may have been influenced by the resources and capacity in the MU and the OU at the time of the birth. Free text comments entered by some reporting midwives indicated that the capacity of both MUs and OUs may have influenced the decision to transfer a woman, or not, following a PPH. Because we were reliant on anonymised data entered directly from medical records, we did not have data on several factors of interest that might have shed more light on this including, for example, staffing levels in the MU and OU around the time of the birth.

This study was planned to run for 12 months but was cut short due to the COVID-19 pandemic. Had data collection been able to proceed as planned, the study would have had greater power as planned to be able to detect associations between PPH and putative risk factors that were uncommon in this population.

## Conclusions

PPH requiring transfer to obstetric care following birth in an MU is a relatively uncommon event in the UK, but incidence may be increasing. The risk factors associated with the most significant increase in the odds of a PPH requiring transfer to obstetric care were a third stage of labour lasting 60 minutes or more, and perineal trauma. Our results about outcomes for women who have a PPH in an MU are broadly reassuring and indicative of appropriate management. It remains important that NHS organisations have robust guidelines about the management of PPH in MUs, appropriate equipment and training, and ready access to transfer when required.

## Supporting information

**S1 Table. Maternal sociodemographic characteristics among women who had a PPH requiring transfer, according to whether they received 'enhanced treatment or care'.** (DOCX)

**S2 Table. Pre-existing clinical characteristics among women who had a PPH requiring transfer, according to whether they received 'enhanced treatment or care'.** (DOCX)

**S3 Table. Clinical characteristics arising during pregnancy among women who received enhanced care and women who did not receive enhanced care.** (DOCX)

**S4 Table. Intrapartum factors among women who received enhanced care and women who did not receive enhanced care.** (DOCX)

**S5 Table. Birth-related factors among women who received enhanced care and women who did not receive enhanced care.** (DOCX)

**S6 Table. Neonatal outcomes among cases and controls.** (DOCX)

**S7 Table. Risk factors for PPH requiring transfer to obstetric care among cases according to the type on unit in which they gave birth.** (DOCX)

**S8 Table. Risk factors for PPH requiring transfer to obstetric care among controls according to the type on unit in which they gave birth.** (DOCX)

**S9 Table. Blood loss among cases and controls by unit type.** (DOCX)

## Acknowledgments

The authors would like to thank all the UKMidSS reporting midwives across the UK who responded to monthly report requests and entered data. We would also like to thank Alessandra Morelli, the UKMidSS Research Midwife, for her role in coordinating data collection, and the management of data queries.

UK Midwifery Study System Steering Group (in addition to RR, JR, CW, DP RP and JJK): Mervi Jokinen (Chair), Royal College of Midwives; Philippa Cox (Vice Chair, from February 2022), Homerton University Hospital; Meena Bhatia (from June 2019), Oxford University Hospitals NHS Foundation Trust; Posy Bidwell, The Royal College of Obstetricians and Gynaecologists (until Feb 2020); Jan Butler (until December 2018), Cambridge University Hospitals NHS Foundation Trust; Karen Joash (from June 2019), Imperial College Healthcare NHS Trust; Eddie Morris (until May 2020), Royal College of Obstetricians and Gynaecologists; Jackie O'Neill (from February 2019), South Eastern Trust; Aung Soe (until October 2021), Medway NHS Foundation Trust; Anna Temke (from February 2019), James Cook University Hospital; Phillis Winter, Midwifery Team Manager at NHS Tayside (until Feb 2019).

## Author Contributions

**Conceptualization:** Rachel Rowe.

**Formal analysis:** Madeline Elkington.

**Funding acquisition:** Jennifer J. Kurinczuk, Dharmintra Pasupathy, Rachel Plachcinski, Jane Rogers, Catherine Williams, Rachel Rowe.

**Investigation:** Rachel Rowe.

**Methodology:** Madeline Elkington, Jennifer J. Kurinczuk, Rachel Rowe.

**Project administration:** Rachel Rowe.

**Writing – original draft:** Madeline Elkington.

**Writing – review & editing:** Jennifer J. Kurinczuk, Dharmintra Pasupathy, Rachel Plachcinski, Jane Rogers, Catherine Williams, Rachel Rowe.

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
