## [Decision Letter · Decision Letter 0]

7 Dec 2022

PONE-D-22-20205Postpartum haemorrhage occurring in UK midwifery units: a national population-based case-control study to investigate incidence, risk factors and outcomesPLOS ONE

Dear Dr. Elkington,

Thank you for submitting your manuscript to PLOS ONE. After careful consideration, we feel that it has merit but does not fully meet PLOS ONE’s publication criteria as it currently stands. Therefore, we invite you to submit a revised version of the manuscript that addresses the points raised during the review process.

We look forward to receiving your revised manuscript.

Kind regards,

Chandan Kumar, Ph.D.

Academic Editor

PLOS ONE

Journal Requirements:

a) Did participants provide their written or verbal informed consent to participate in this study?

5. One of the noted authors is a group or consortium “UKMidSS Steering Group”. In addition to naming the author group, please list the individual authors and affiliations within this group in the acknowledgments section of your manuscript. Please also indicate clearly a lead author for this group along with a contact email address.

Reviewers' comments:

Reviewer's Responses to Questions

**Comments to the Author**

1. Is the manuscript technically sound, and do the data support the conclusions?

Reviewer #1: Yes

Reviewer #2: Yes

2. Has the statistical analysis been performed appropriately and rigorously? 

Reviewer #1: I Don't Know

Reviewer #2: Yes

3. Have the authors made all data underlying the findings in their manuscript fully available?

Reviewer #1: No

Reviewer #2: Yes

4. Is the manuscript presented in an intelligible fashion and written in standard English?

Reviewer #1: Yes

Reviewer #2: Yes

5. Review Comments to the Author

Reviewer #1: • This is a prospective national (England UK) population-based case-control study in all MUs in the UK using 27 the UK Midwifery Study System (UKMidSS). This has become a highly successful method of capturing prospective data from maternity settings in the UK and has informed practice relating to many different obstetric complications. There is a good series of highly referenced papers that have previously published using this methodology with high impact. Previously published PPH papers from UKOSS have been very informative and this paper would add to this series of publications.

• The study was supposed to run over 12 months based on the incidence of PPH transfers from midwifery to obstetric settings from the literature but was stopped at 6 months because of the COVID pandemic. This may have led to some reduction of impact relating to some of the outcomes/recommendations but as the incidence of transfer was higher than previously reported the study seemed adequately powered.

• Overall, I found the writing style quite long with the frequent use of the pronoun “we”. I feel that the paper would benefit from a more focused writing style. Consideration should be made to removing “we…. did this” to “……… was undertaken” or something similar.

• As journals are international I think there needs to be a clear but concise explanation of how and why the UK maternity system is set up with MLC/CLU . It is also somewhat confusing in the text as clearly some high-risk women were admitted to the MLU which is clearly contrary to national guidance but can be the result of local practice or personal preference. Again, this requires brief explanation.

• In the discussion there needs to be clearly written highlights to what this paper adds to the literature. I think there are many.

• I would also like to see more of an acknowledgement that a PPH rate of 3% in a low-risk population is significant and not without maternal morbidity. One lady had a 5L PPH. In addition, the rate appears to be higher than previously reported. Part of the conclusion must therefore to maintain safe care there should be robust guidelines on MLU management of PPH and robust rapid transfer policies. I found the tone too reassuring.

Reviewer #2: This paper presents a contemporaneous snapshot of transfers form MUs following or for treatment for PPH. As such this is novel as these data have not previously been reported (as identified BirthPlace reported all transfers). The high response rate from units was impressive and clearly identities the effectiveness of the UKMidSS methodology. Selection of the preceding birth as a control, with no adjustment, for example, for parity was interesting, and could have been better rationalised - ie, by definition these are all 'low risk' women, and may have contributed to the imbalance of primiparous women included in the control group. Especially given the general advice that MU care is associated with fewer interventions and better outcomes for multiparous women.

The "risk profile' of this cohort is interesting and some rationale or explanation around this might have proved useful. For example many AMUs and FMUs have guidelines excluding women with high BMIs, or pre-labour complications and yet there were several included in this cohort.

Table 2 I calculate the incidence related to previous PPH in cases as 9.8% not 10.2 as reported. This is still strongly significant but changes the OR slightly (2.74 95%CI 1.81 to 4.10). I note the numbers reported in Table 4 as adjusted OR are more similar.

Table 3: Indicates 84 cases and 56 controls had labour induced. Is there any explanation around this? Given that induction of labour is a medical procedure and initiation is outside the midwife's scope of practice.

Similarly there were (a few) women (total n= 54) with complications identified before labour, more than twice as many of these were in the cases rather than the controls. Similarly there were 77 incidences where fetal complications were identified before labor, and again more of these were in the case group than the control (1.9% versus 3.2%). Similar disparities were seen in complications identified during labour but there are no references to these in the discussion. The authors reported water immersion in labour was a significant risk factor for PPH, but this is not expanded in the discussion. Given that many women choose birthing in MUs in order to use water immersion for labour and/or birth , this could be an important message about risk factors.

The incidence of instrumental vaginal births in these low risk settings is explained in the discussion.

The incidence of 3rd and 4th degree tears appears high, and although contributory to transfer for PPH, these women are more likely transferred for repair .

Was the higher incidence at later gestation due to these women being induced? As this is a MU study one would assume all women were managed expectantly but the number of IOLs suggests otherwise, and were some additional women augmented, which has not been documented? Given the known associations between syntocinon use in the 1st and 2nd stage of labour and subsequent PPH, this would not be surprising.

It is a shame this study was stopped prematurely, as I am sure some of these issues are down to smaller numbers than anticipated, although almost 40000 women is not a small number!

DISCUSSION: This appears to omit any comments about the profile of women birthing in MUs. There are several in this cohort that would appear to be higher risk than low risk and therefore some discussion around choice and availability should be included. Note that the authors do state that transfer was influenced by staffing levels.

I think there are also considerations around workforce skills and the impact that plays in transfer, and care provided and these are not alluded to in the discussion.

6. PLOS authors have the option to publish the peer review history of their article (what does this mean?). If published, this will include your full peer review and any attached files.

Reviewer #1: **Yes: **Rachel Collis

Reviewer #2: No

---

## [Decision Letter · Decision Letter 1]

7 Sep 2023

Postpartum haemorrhage occurring in UK midwifery units: a national population-based case-control study to investigate incidence, risk factors and outcomes

PONE-D-22-20205R1

Dear Dr. Elkington,

We’re pleased to inform you that your manuscript has been judged scientifically suitable for publication and will be formally accepted for publication once it meets all outstanding technical requirements.

Kind regards,

Ahmed Mohamed Maged, MD

Academic Editor

PLOS ONE

Additional Editor Comments (optional):

Reviewers' comments:

Reviewer's Responses to Questions

**Comments to the Author**

1. If the authors have adequately addressed your comments raised in a previous round of review and you feel that this manuscript is now acceptable for publication, you may indicate that here to bypass the “Comments to the Author” section, enter your conflict of interest statement in the “Confidential to Editor” section, and submit your "Accept" recommendation.

Reviewer #3: (No Response)

Reviewer #4: All comments have been addressed

2. Is the manuscript technically sound, and do the data support the conclusions?

Reviewer #3: Partly

Reviewer #4: Yes

3. Has the statistical analysis been performed appropriately and rigorously? 

Reviewer #3: I Don't Know

Reviewer #4: Yes

4. Have the authors made all data underlying the findings in their manuscript fully available?

Reviewer #3: No

Reviewer #4: Yes

5. Is the manuscript presented in an intelligible fashion and written in standard English?

Reviewer #3: Yes

Reviewer #4: Yes

6. Review Comments to the Author

Reviewer #3: * This manuscript highlight the on the Postpartum haemorrhage occurring in UK midwifery units: a national population-based case-control study to investigate incidence, risk factors and outcomes.

* I see the manuscript is technically partly sound and the claims are convincing and supported by the collected data. But, there are small numbers and missing data due to the reasons of privacy ????

* Moreover, the authors were reliant on anonymised data entered directly from medical records. In addition, the authors should go through the journal guide lines and read it carefully and correct all statements and declarations as per journal instructions.

* The authors have to provide a good discussion of the obtained results.

Reviewer #4: The authors investigate the incidence, risk factors and outcomes of national population-based case-control in UK midwifery units. As per the revised contributions and the present form of manuscript is recommended.

7. PLOS authors have the option to publish the peer review history of their article (what does this mean?). If published, this will include your full peer review and any attached files.

Reviewer #3: No

Reviewer #4: **Yes: **Dr Goutham Kumar Nadakuditi

---

## [Editor Report · Acceptance letter]

25 Sep 2023

PONE-D-22-20205R1 

Postpartum haemorrhage occurring in UK midwifery units: a national population-based case-control study to investigate incidence, risk factors and outcomes 

Dear Dr. Elkington:

I'm pleased to inform you that your manuscript has been deemed suitable for publication in PLOS ONE. Congratulations! Your manuscript is now with our production department. 

Kind regards, 

on behalf of

Professor Ahmed Mohamed Maged 

Academic Editor

PLOS ONE